

# From Five to Thirty-Five: Fostering the Next Generation of Arctic Scientists

Jenny V. Turton[1], Naima El bani Altuna[2], Charlotte Weber[3], Salve Dahle[3], Nina Boine Olsen[4], Elise Fosshaug[4], Katrine Opheim[1,5], Julia Morales-Aguirre[1], and Astrid Wara[4]

[1] Arctic Frontiers, Tromsø, Norway

[2] North Atlantic Marine Mammal Commission (NAMMCO), Tromsø, Norway

[3] Akvaplan-niva, Tromsø, Norway

[4] Science Centre of Northern Norway, Tromsø, Norway

[5] UiT – The Arctic University of Norway, Tromsø

*Correspondence to*: Jenny V. Turton (jenny@arcticfrontiers.com)

**Abstract.** Out-of-classroom Education and Outreach (E&O) initiatives can improve the uptake of Science, Technology, Engineering and Maths (STEM) courses at higher education and can help address gender balances within the fields. Arctic Frontiers, a non-profit organisation based in Tromsø, Norway, has been running various projects under the Young Program banner since 2012. Through their four programs, ranging in levels from Kindergarten to Early Career Professionals (and ages

from 5 to 35), over 3000 individuals have been exposed to Arctic research and science through workshops, mentoring, career seminars and excursions. With the rate of climate change in the Arctic and the geopolitical changes in the region, E&O initiatives focusing on Arctic science are now even more crucial, but potentially more challenging to run. This study outlines the main educational activities and the best practices from the last decade, to provide a template for science communication and outreach. Additionally, a first analysis of the reach and success of the program is provided, by identifying trends in

participant numbers, geographical interest and demographic identifiers.

## 1 Introduction

At both the education and career level, Science, Technology, Engineering and Maths (STEM) subjects are typically still characterised by an imbalance in gender and lack of minority groups across the globe (Gibney, 2016). Greater numbers of skilled and educated people (especially women) in STEM fields will benefit many nations who are struggling to find skilled

workers, as well as increasing innovative potential and readdressing gender imbalances (Bøe et al. 2011).

The Arctic nations are no exception. In Norway, where more women are studying at university than men, only 30% of full professor positions are held by women (Lekve and Gunnes, 2022). In Canada, only 4% of Indigenous Peoples pursue an education in STEM (Cole et al. 2022). Therefore, there are still large inequalities in access to STEM education for minority groups, and a lack of support for women continuing their careers in academia (Pew Research Center, 2021).

In the Arctic and northern regions of the Arctic nations, there is an additional issue facing STEM fields. Out-migration of skilled workers to southern urban centres and capital cities (all located outside of the Arctic Circle) is a common trend across



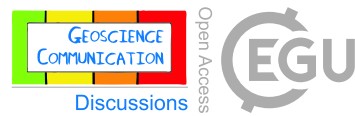

the Arctic (Nilsson and Larsen, 2020). We observe that Arctic science has a double 'Leaky Pipeline' problem. The Leaky Pipeline is a common term to refer to individuals from under-represented communities who journey away from STEM education or careers due to pressures and inequalities in the field (Cole et al. 2022). In the Arctic, there is the additional

leakage of highly skilled, young (often minoritised genders and ethnicities) people to the south. Therefore, it is crucial that young people residing in the north and the Arctic can develop an interest for, and skills in, STEM subjects, which make them highly employable and invested in remaining in the north.

In studies in the USA and Europe, it has been documented that out-of-school outreach projects lead to an increase in students' motivation to continue with STEM education and enhance their appreciation for STEM in real-life applications

(Vennix et al. 2018). Consistent within the literature, is the knowledge that mentors and role models that reflect the students' identity (here, early-career, Indigenous or local northern residents), are key for increasing equality in the STEM fields (Krikorian et al. 2020; Cole et al. 2022). In Alaska, museums have been highlighted as crucial STEM outreach facilitators for Arctic youth. The retention of teachers in Alaska is poor: 60% of Alaska's teachers leave the Arctic after just two years of teaching (Anderson et al. 2017). Out-of-school outreach provides informal learning and, when combined with industry visits

or interactions, students feel more relaxed about learning complex subjects (Vennix et al. 2017).

Scientists and researchers are now encouraged or forced to engage in science communication and outreach as part of the requirements of funding agencies, due to the use of public money. However, non-academic engagements are often transient, ad-hoc or unsuccessful (Peters et al. 2024). Even scientists who actively highlight the importance and benefits of outreach and science communication for both the public and the science, are unlikely to engage in non-academic dissemination

(Roberts et al. 2009). The burden and additional work for scientists to propose and organise their own outreach and science communication is large. This highlights the need for other organisations working at the forefront of science and local communities to provide the structure for scientists to engage with the public.

Whilst the broad scope of Arctic science includes not only STEM subjects, but also social sciences, anthropology, sociology and politics, the focus of the Education and Outreach (E&O) programmes run by Arctic Frontiers is often on the physical

and environmental sciences, especially climate and oceans. Arctic Frontiers is a non-profit organisation based in Tromsø, Norway and owned by Akvaplan-niva, an aquaculture and environmental research and consultancy company. Arctic Frontiers aims to be a catalyst for decision-making and network-building within the Arctic and has over 20 'partners' which include many universities and research institutes in Norway, as well as the Norwegian Polar Institute, and private companies with an interest in the Arctic. Since 2007, Arctic Frontiers has been gathering Arctic stakeholders from science, business,

politics, and local communities for discussions regarding a range of Arctic-relevant themes, such as climate change, geopolitics, energy transition and demographic changes. A number of E&O projects, aimed at all ages between five and thirty-five, have been developed and run both nationally (in Norway) and internationally. With at least 10 years of experience in developing and implementing E&O activities, and both qualitative and quantitative feedback, four key projects now feature in the annual running of Arctic Frontiers.



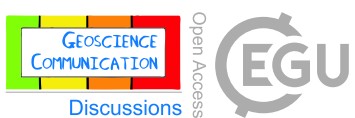

The aims of running a comprehensive 'Young Program' (as it is broadly termed at the company) are four-fold: 1) To encourage a greater number of students to take further education in STEM subjects, 2) to highlight opportunities for STEM careers within the Arctic, 3) to inspire generations of Arctic enthusiasts, who will go into science, research, policy-making and sustainable industry careers, and 4) to engage scientists in regular, impactful science communication with the public.

The aim of the paper is to highlight the importance of continued and sustained E&O projects at all stages of education and
early career development, to ensure interested and inspired scientists remain working on, and living in, the Arctic. The first section of the paper describes the various programs, to create a template and best practices for others wishing to engage in E&O programs, especially with harder-to-reach communities in the rural Arctic. The second part of the paper investigates the impact and reach of the four programs over the last decade, through analysis of the participants and showcasing the qualitative feedback from participants.

**2 Program Descriptions**

The various young programs at Arctic Frontiers were developed since the beginning of the company in 2007 (Dahle et al. 2019), however funding difficulties meant that the activities were more consistent and structured after 2012, forming a complete Young Program. There are four key projects which occur annually as part of the Arctic Frontiers Young portfolio (Figure 1): Science for Kids, Science for Schools, Student Forum and Emerging Leaders. These programs have been
developed and improved over the years, and interannual changes reflect funding priorities, geopolitical circumstances, and evaluation of the program. The details of the standard programs are provided in detail below, however larger changes have occurred due to the COVID-19 pandemic and the Russian invasion of Ukraine. Therefore, between 2020 and 2024, some of the activities were suspended or altered due to the extenuating circumstances. Additionally, some information has been lost due to lack of consistent archiving practices prior to 2020, and GDPR regulation changes in 2018. Since 2021, a full
archiving and reporting system is in place for competent monitoring of projects.

All four projects within the Young Program are based on scientific knowledge and cognitive understanding as an educational approach to learning about the Arctic. This is the most common basis for climate change educational programs in a study of 220 publications focusing on E&O projects (Rousell & Cutter-Mackenzie-Knowles, 2019). The second most common educational approach is curriculum and pedagogy, which is the basis of the two programs with the youngest target audience
at Arctic Frontiers (Table 1).



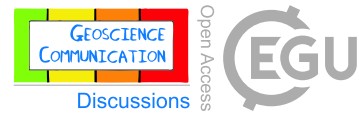

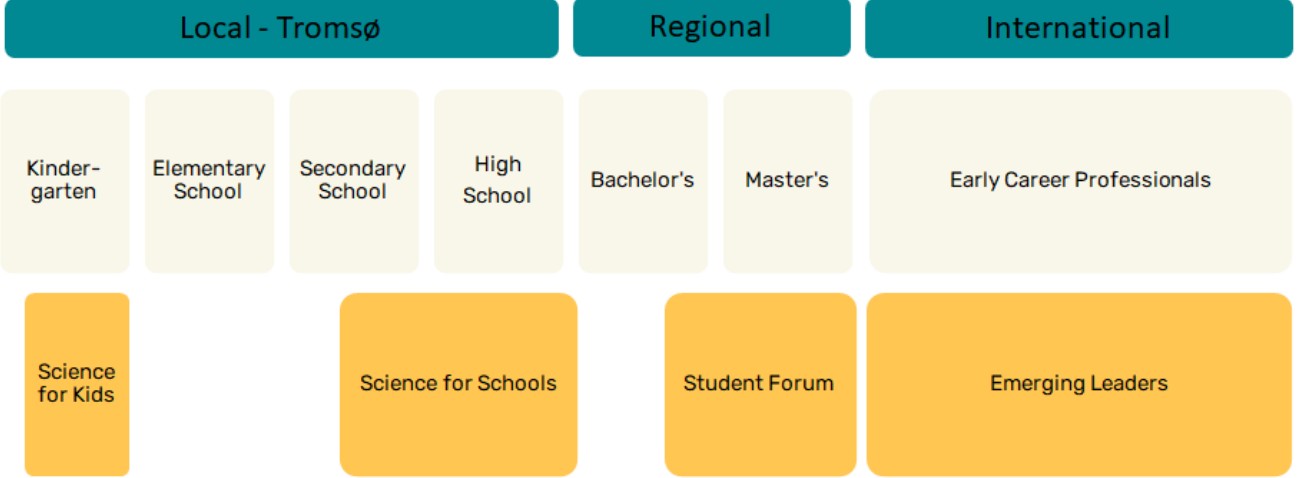

**Figure 1: Arctic Frontiers Young portfolio programs with respect to educational and professional development stages. Early career professionals includes PhD candidates and those in professional employment in industry, policy or public office.**

## 2.1 Science For Kids

'Science for Kids' is the early-years project for children in the earliest years of formal education (here termed kindergarten for international translation), between the ages of four and five. This project is currently run in collaboration with the Science Centre of Northern Norway (Nordnorsk vitensenteret) and the Arctic University of Norway, Tromsø (UiT). This is the newest activity in the E&O portfolio, and it began in 2017 as a pilot, and as a full program in 2018 with seven kindergartens. The project is offered to new kindergartens each year, to ensure the spread of the didactic thinking and working methods in the project. The aim of this project is to initiate an interest in science, create positive associations with science, and to encourage the children to question the world in which they live. Furthermore, it is a goal to encourage and give confidence to the kindergarten teachers in running inquiry-based projects and to educate and support them in building experience on how to guide children based on their innate curiosity and desire for knowledge.

The project begins with Inspiration Days, where children visit the science centre to hear about how scientists study the ocean, atmosphere and ecosystem. This is an inquiry-based class, where the children's curiosity and exploration are encouraged. The children are given time to ask questions and to participate in play-based activities and experiments. The kindergarten teachers are then invited to the Science Centre of Northern Norway for a course, where they are given skills, information and resources to lead the children in inquiry-based exploration about Arctic science. After a period of investigations carried out by the children, the Science Centre arranges a meeting with an expert on the topic, to provide support for the teachers in the kindergarten, and trigger further curiosity in the children. Some of the research is summarized on a poster, including the research question, hypotheses and conclusions. The children showcase their posters in a 'poster festival' , which concludes with a science show from centre staff.



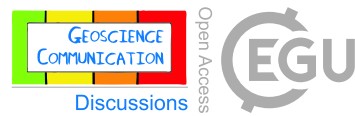

## 2.2 Science For Schools

The 'Science for Schools' project is aimed at secondary and high school level (Figure 1). The project, which has been running since 2014 is organised in connection with the Science Centre of Northern Norway, and presently eight schools in Tromsø. This E&O project aims to show pupils the steps of research, including deciding a hypothesis, conducting experiments and showcasing results via a conference. There is a three-step process: Inspiration Days, Project Implementation, Science Conference.

'Inspiration Days' take place in early October, when  the pupils listen to scientists give an engaging talk about their research, field or lab work, and career development. Scientists are invited by Arctic Frontiers to participate, and care is taken to ensure gender parity amongst the speakers, as well as including early career professionals, a broad range of science themes and other areas of inclusion. The language of speakers is either Norwegian or English, depending on what the school prefers. The main aim is to initiate thought about which theme the pupils would like to investigate when they start their science project. For example, in 2022, speakers included marine biologists, meteorologists, aquaculture specialists, glaciologists and oceanographers.

The second step of the project is a group research project, whereby the pupils devise a hypothesis related to Arctic science (very broad in order for the pupils to investigate their main interests) and decide on a method or experiment to complete the research project in  three months, working in small groups. Many pupils reach out to institutes for support with their project and conduct lab experiments or fieldwork. To conclude, they create an A0 poster of their project, results, and conclusions in January. The teachers select a number of groups to attend the third activity based off the strength of their project; typically three groups are sent from every school to the next stage.

The third stage of the project is a three-day science conference, hosted at the Science Centre of Northern Norway, for the students to listen to oral talks from practicing scientists and PhD candidates, and they present their posters to a number of judges (Figure 2).  The judges are often master's students or PhD Candidates, so that they have the experience of evaluating scientific posters and providing constructive feedback to students (Figure 2). In the last few years, the Association of Polar Early Career Scientists (APECS), have been collaborators. The format replicates an actual science conference, where oral talks, poster presentations and awards for presentations are provided.


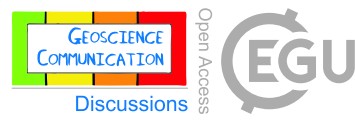

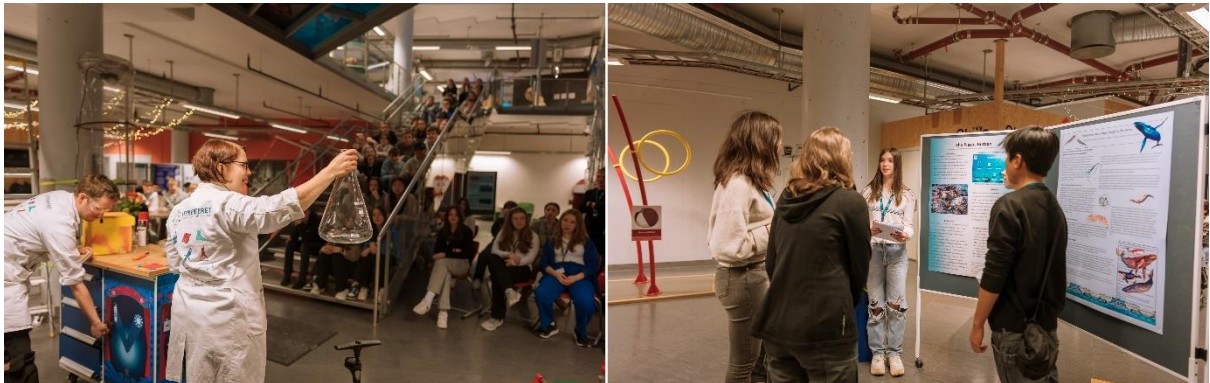


**Figure 2: High school students visit the Science Centre of Northern Norway and interact with scientists and educators (left). The school pupils present their posters to the judges during the Science For Schools program (right).**

There are nine awarded posters from the science conference for best visual poster, best scientific content and best overall. The awardees are invited to present their poster at the Arctic Frontiers annual conference, as part of the science poster

session. This allows the students to experience a scientific conference and interact with scientists. In 2022, pupils were also given an additional opportunity to visit the Kronprins Haakon Ice breaking research vessel, owned and operated by the Norwegian Polar Institute, Institute of Marine Research and UiT – The Arctic University of Norway.  Many of the topics chosen by the pupils focus on plastic pollution in the ocean, problems associated with commercialised fishing (e.g salmon lice), and visible climate change impacts. These reflect not only the key Arctic talking points in Northern Norway, but also

the collaborating institutes in the project.

Some schools have communicated the importance of this project in their annual curriculum, and pupils from these schools present their research each year. For other schools, attendance is dependent on the availability of teachers and the interest of pupils in extra-curricular groups such as science and technology groups.

**2.3 Student Forum**

The Arctic Frontiers Student Forum (AFSF) gathers Bachelor and Master students from northern Norwegian and Finnish universities for one week in Tromsø, to build a cross-border and cross-sectorial learning experience and networking arena. The group usually consists of approximately 15-20 students and the project has been running since 2016. Applicants are chosen based on the loose criteria that they must be students with an interest in learning more about the Arctic and sharing their Arctic experiences. Considerations for gender-balance and geographic diversity are also taken into account. This

program has  experienced the largest changes over the last 8 years, due largely to geopolitical changes in the Arctic, and to better reflect the needs of the students.

There are four aspects to the AFSF: 1) Workshops for informal learning, 2) Social and cultural program, 3) Group project, 4) Mentoring.



The theme of the informal learning workshops varies by year but topics have previously included the role of the Arctic Council, sustainability in the Arctic, importance of science diplomacy, ocean preservation and management, mental health in the Arctic (with invited professional psychologists) and climate change mitigation.

The second component is a social and cultural program. As the AFSF runs alongside the Arctic Frontiers annual conference, many of the social activities are already organised, such as art exhibitions, youth receptions and Pecha Kucha science

communication evenings. Combining with the main conference activities provides greater networking opportunities and exposes the students to key Arctic stakeholders.

The group project sees the students divided into interdisciplinary groups, with interests ranging from business and economics to physical sciences, law, and political sciences. Students are asked to come up with a question which they will explore throughout the week and create a proposal for project funding to address the question. The project is to align with the general

themes of the Arctic Frontiers conference: geopolitics in the Arctic, sustainable ocean development, climate change impacts, and the future of the Arctic Council.

The mentoring part of the forum has two aspects: case guidance mentoring and career mentoring. The former should support the group projects, where mentors with experience in the theme are assigned to a group to guide the students with their theme, provide contacts for them to reach out to and suggest relevant sessions to attend for more information. The career

mentoring aspect is more traditional mentoring, where students receive guidance on career options. The AFSF participants and mentors are now connected afterwards via a LinkedIn group and are invited to a digital career seminar.

Since the beginning of the project, the shift from lecture-based to informal learning has taken place. The student forum now places emphasis on peer-to-peer learning, independent goal setting and projects. The reason for this transition is two-fold: Firstly, through feedback and evaluation of participants over the years, it has become clear that students wanted the ability to

learn from each other. Secondly, informal and peer learning techniques have been shown to promote interest and involvement of students notably in STEM fields (Goff et al., 2019; Roberts et al., 2018).

Prior to 2022, the project gathered participants from northern Norway and northwest Russia with funding from the Norwegian Barents Secretariat. Due to Russia's invasion of Ukraine, the Barents Secretariat has halted their funding programs and travel restrictions make travel impossible. In 2022, the AFSF did not operate and in 2023, the AFSF returned

with a Nordic focus.

## 2.4 Emerging Leaders

Emerging Leaders is more of a mentoring and career development project than strictly E&O, however there are many aspects of this which overlap with E&O and also align with the aims of Arctic Frontiers to support Arctic scientists and professionals throughout the beginning of their career. The Emerging Leaders program evolved out of a former project called 'The Young

Scientist Forum', run in connection with the Arctic Marine Ecosystem Research Network (ARCTOS), which took place in



2007. However, from 2012 onwards, these became two separate events. The description below relates to the most structured and consistent phase of the program.

The Emerging Leaders program has been running since 2012 and sees approximately 30 PhD Candidates and Early Career Professionals each year travel through northern Norway in January and attend the Arctic Frontiers annual conference. This is

a truly international program, with over 15 countries (nationalities) participating in 2023.

Participants must apply for the program in September, and a team of three evaluates the applications and selects candidates, ensuring a diverse range of participants in terms of age, gender, nationality, profession and/or research area. The only strict criterion is that the participant must be between 18 and 35 years old.

The selected participants meet in Bodø, northern Norway, in late January and embark on a 5-day travel and learning

excursion via boat and bus across 530 kilometres to Tromsø (Figure 3). Each day, participants meet a number of local business leaders, scientists and other professionals, for workshops, lectures and mentoring meetings. Additionally, participants are encouraged to engage in peer-to-peer learning on relevant topics, due to the high level of expertise in the group. Since 2022, there has also been an alumni program developed, to allow the Emerging Leader participants to keep in touch with their peers and previous cohorts via private social media groups. Additionally, leadership workshops are held

twice a year for alumni, and digital meetups are provided and opportunities for speaking arrangements and participation at other events are advertised.

During the COVID19 pandemic, the 2021 Emerging Leaders program was run online with 25 participants being chosen from the previous 5 annual cohorts. The participants were divided into two groups focusing on climate change and the pandemic. The goal was to develop lessons learned from both crises to deliver at the Arctic Frontiers 2021 conference which was also

digital.

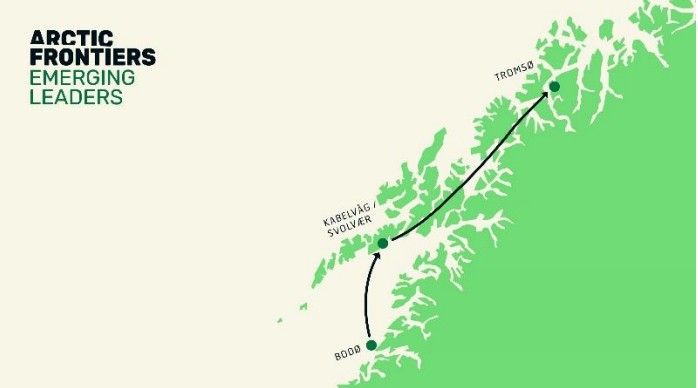

**Figure 3: Graphical representation of the Emerging Leaders route from Bodø to Tromsø in Northern Norway. Graphic created by Reibo AS. This figure is taken from www.arcticfrontiers.com/emergingleaders/emerging-leaders.**


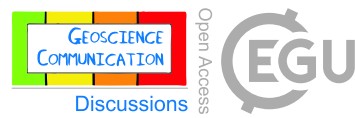

**Table 1: Summary of the Young Program components and the approaches taken for public E&O. The Educational Approach is based on the categories by Rousell and Cutter-Mackenzie-Knowles (2019).**

| Program | Audience | Key Activities | Educational Approach |
|---|---|---|---|
| **Science For Kids** | Kindergarten children (approximately 4-5 years old) and Kindergarten teachers in Tromsø. | Poster party with science experiments, kindergarten teacher workshops. | Curriculum/pedagogy, experimental/participatory, child-framed, knowledge-based/cognitive. |
| **Science For Schools** | Secondary and high school students (approximately 14-17 years old) in Tromsø. | Inspiration days, 3-month research project, poster presentation at youth science conference, selected few attends Arctic Frontiers science poster session. | Curriculum/pedagogy, experimental/participatory, child-framed, knowledge-based/cognitive. |
| **Student Forum** | Bachelor and Masters students (approximately 20-26) from Norway and Finland. * | Workshops for informal learning, social and cultural program, group project, mentoring, attending Arctic Frontiers annual conference. | Public communication, knowledge-based/cognitive, experimental/participatory. |
| **Emerging Leaders** | Early Career Professionals and PhD Candidates (upper age limit of 35) from international backgrounds. | Digital leadership workshops, alumni network, career mentoring, business and science talks, attending Arctic Frontiers annual conference. | Public communication, ethical/philosophical/critical, knowledge-based/cognitive, experimental/participatory. |

*2022-2024 program was focused on students from Northern Norway and Finland, but this is dependent on funding.

## 225  3 Impact and Evaluation

The results presented here summarise the key performance indicators used for assessing the success of the E&O projects and tracking development in terms of accessibility and diversity. Additionally, we assess the impact of the programs through growth in the number of participants, and their feedback.

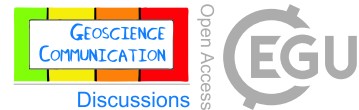

### 3.3 Science for Kids & Science for Schools

In its latest phase (2023-2024), children from five kindergartens in Tromsø are gathered at the Science Centre as part of the Science For Kids project (Figure 4). This project has seen a variable number of children participating (from 83 to 235). The number of kindergartens involved has fluctuated between five and thirteen since the program started in 2018. In total, over 550 children and 90 kindergarten teachers have participated in the project.

In the last three years, over 400 pupils from Tromsø have joined the Science For Schools Inspiration days from 5-8 different schools (Figure 5). In most years, information on the number of pupils is available but not the number of schools, as students from smaller schools are combined on a number of the days. However, there has been an increase in the number of pupils and schools, which has largely been attributed to greater visibility of the event within the local newspapers, broader networks of those involved in organising the event, and more consistent funding.

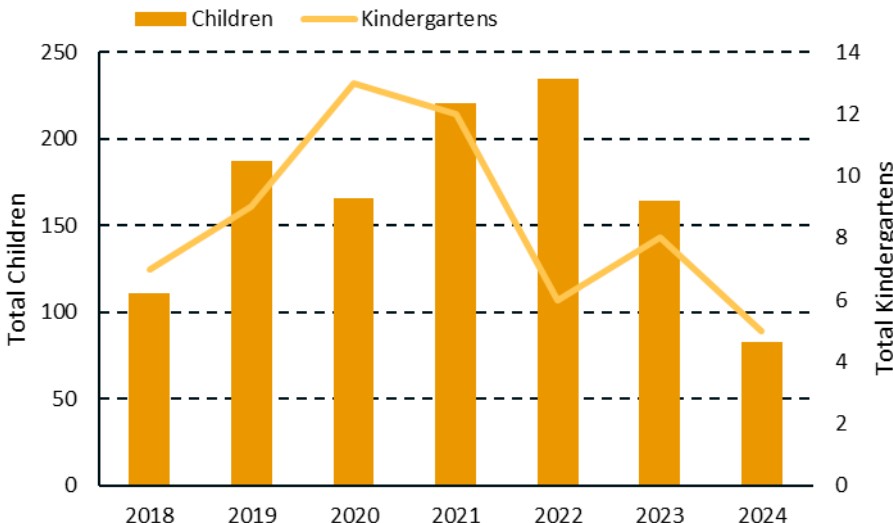


**Figure 4: The total number of participants in the Science for Kids program each year and the number of participating local kindergartens Null years represent data gaps as explained above. 2021 and 2022 estimates of participating children are underestimates taken from video and picture evidence of the events.**

Data availability is poorest for both the Science for Schools and Science for Kids programs, as the project is a collaborative
effort between the schools, kindergartens, the Science Centre of Northern Norway and Arctic Frontiers, with key information known by the teachers. Demographic data on the children are not collected (e.g gender) and some information on the number of schools and kindergartens is unknown (Figure 4 and 5). Due to the high number of children participating, it is difficult to track their further development or trajectory within STEM subjects, however qualitative discussions with teachers and students have revealed that many children are inspired by the program and do decide to take STEM subjects at university
level (Table 2).

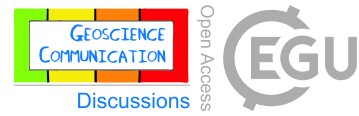

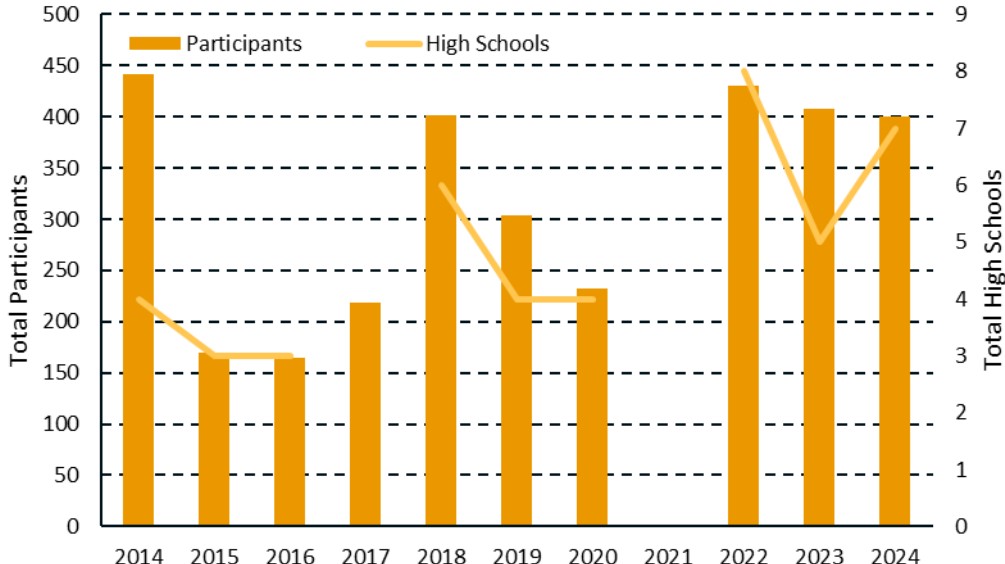

**Figure 5: The total number of participants in the Science for Schools program each year and the number of participating local high schools. Null years represent data gaps as explained above.**

**Table 2: Quotes from the participating kindergarten teachers in Science for Kids and high school pupils in Science for Schools. These quotes are translated from Norwegian into English, and therefore not direct quotes.**

| Science for Kids – Kindergarten Teachers |
|---|
| The project has helped to start philosophical conversations with children and has helped us to ask more questions together with the children. |
| The children have expressed that it has been both educational and fun. They have been involved in many different things and learned a lot. They now ask and inquire about how things are and work. |
| I left the course with renewed vigour and feeling very inspired. I learned more about the theory around scientific method, concrete tips on how to let the children lead a research project. I have a better understanding of research and how to carry it out in the best possible way in kindergartens. |
| We have included the children's thoughts and input in a different way than when we adults set up a problem for them. What should we research? It was the children who decided. |
| **Science for Schools – Participants** |
| I liked that you had to think quite a lot for yourself, and that you had to find a problem and do a project on it. It gave me an insight into what real science work is. |
| Good way to acquire knowledge as we got to try and see what it's like to do research. We got the opportunities to work interdisciplinary and that we get time to immerse ourselves in a topic. |
| It was a variation on our school work and we had opportunities to work together with researchers and organisations. |

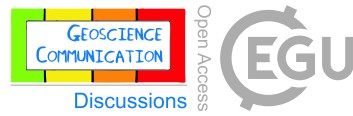

### 3.2 Student Forum

The student forum program is more reliant on the success of short-term project funding than the other E&O activities of
Arctic Frontiers. The number of participants and the participating universities also depends on the focus of the granted
funding. In more recent years, the number of students each year has decreased (see Figure 6), as the program has become
more intensive, taken on a more structured format (see Program Descriptions), is now better connected with the Arctic
Frontiers annual conference, and has received reduced funding. The international aspect of the program has remained
though, with over half of the students affiliated to universities outside of Norway (Figure 6).

Between 2016 and 2023, the student forum has connected over 140 students from across Russia and the Nordic countries,
and provided mentoring, career development opportunities and out-of-classroom learning experiences. Whilst specific
demographic data are not collected, Indigenous peoples have been involved in the Student Forum for at least the last two
years, through the inclusion of the Sami Applied Sciences University.

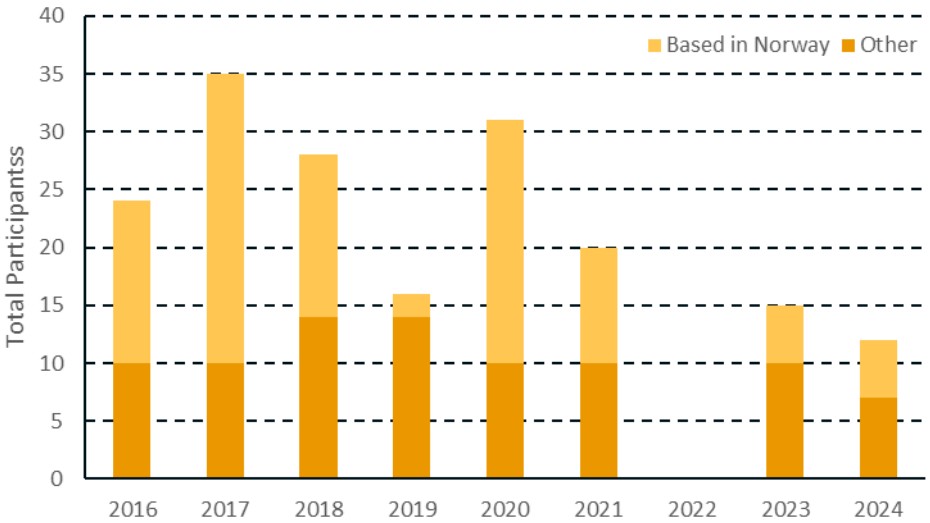


**Figure 6: The total number of participants in the Student Forum program each year divided by origin. Note that the program was note held in 2022 due to the suspension of financial collaboration between Norway and Russian institutes. In 2019, the program was adapted under the Barents Young Programme which was largely supported by Russian institutions.**

Qualitatively, students report positive experiences and regularly use phrases such as 'engaging', 'eye-opening to see different
perspectives', and 'opportunities to discuss with people of different backgrounds' (Table 3).

A growing challenge facing the Student Forum and similar events is the inability to provide a long-term approach, due to the
requirement of continued funding applications. The majority of project funding is limited to two to three years. In Tromsø,
there is the added challenge that many E&O events are organised by similar organisations and UiT, which can reduce the



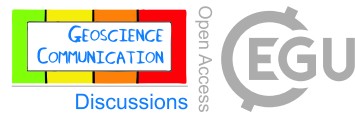

number of applications from students based in northern Norway. An enhanced marketing and communications campaign

since 2022 has seen increased awareness of activities in other areas of northern Norway.

### 3.1 Emerging Leaders

Since 2012, when the Emerging Leader program began, the annual cohort has increased to a maximum of 33 in 2023. The

total participant numbers, as well as gender balance and Arctic vs non-Arctic country split are presented in Figure 7. In 2012

and 2013, records were incomplete, with information on participant gender and affiliation for just 9 people in each year

(Figure 7a). The number of participants identifying as women has fluctuated between 41% in 2014 and 74% in 2022. Both

the number of applicants and selected participants has been dominated by people identifying as women since 2015. The

higher percentage of participants identifying as women is a reflection of the significantly higher number of women than men

applying to the program. The selection criteria have varied over the years, but foundationally consider several factors

including interest in Arctic issues, professional status (e.g PhD candidate, business leader) and age. Additionally, we strive to

have diverse participants in terms of gender, country of residence (Figure 8), profession and when all other factors are

similar, prioritising Indigenous Peoples and Arctic residents.


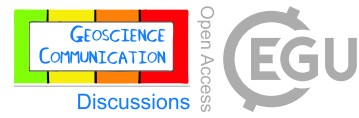

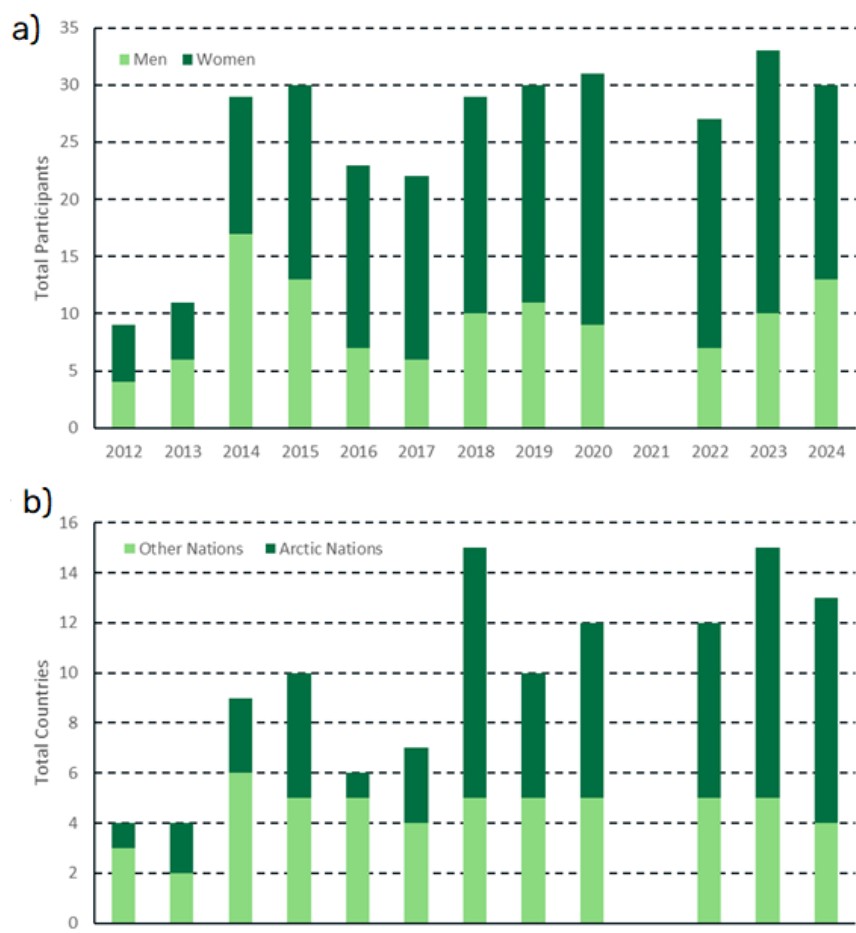

**Figure 7: a) The total number of participants in the Emerging Leaders program each year, split by gender. Gender was in some cases assumed from name and photograph, as gender information was not collected for all years. b) The total number of countries represented in each Emerging Leader program (where the individual is residing, not their nationality), split into non-Arctic and Arctic countries. In 2021, the program was run online with participants from the past 5 years due to COVID19 restrictions.**

The number of Indigenous peoples participating in Emerging Leaders has only been counted in the most recent years. Since 2022, Indigenous participants represented 12-13% of the Emerging Leaders cohort. Whilst Arctic relevance is the main criteria for attending the Emerging Leaders program, Figures 4b and 5 highlight the interest of non-Arctic nations in attending the program. The majority of those from non-Arctic nations are PhD candidates or researchers who are focusing on the Arctic in their research. The percentage of non-Arctic nations attending each year has decreased relative to Arctic nations. This is partly due to the funding for approximately 15 spaces (roughly half of the annual cohort) being ring-fenced for those from Canada and Norway. The total number of attendees each year has been between 25 and 30 for the last 7 years (excluding 2021 due to COVID19). This is seen as the maximum size which allows for good connections within the group and logistically possible for travel along the remote Arctic coast.


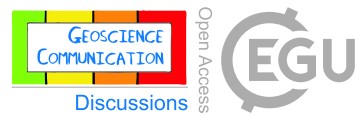

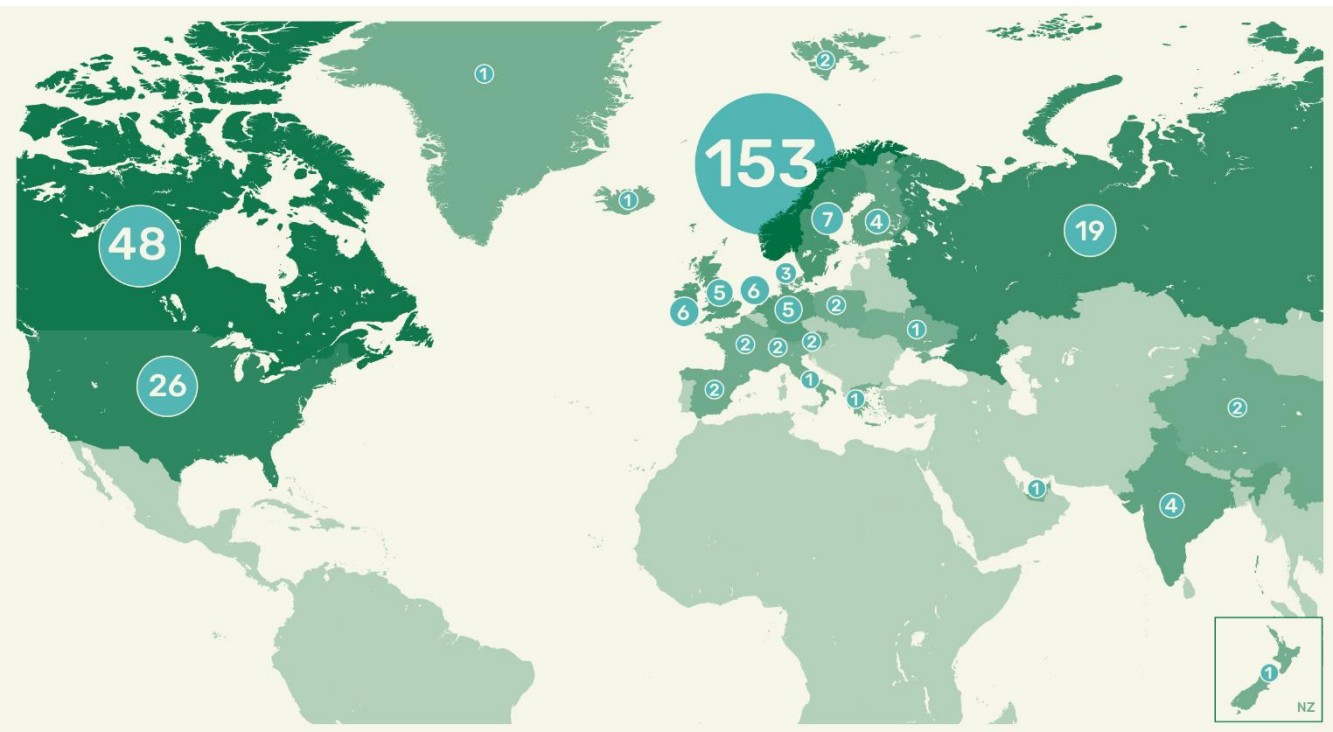

**Figure 8: Graphical representation of the country of residence of all known Emerging Leaders between 2012 and 2024 (location when they applied for the program). Figure created by Reibo AS graphic design company.**

An enriching aspect of the Emerging Leaders program is the diverse employment sector of participants (Figure 9). Whilst those in academia and research (including PhD candidates) dominate the annual cohort in all years since 2014, the number of participants who work in policy and business sectors has been increasing since 2020 (Figure 9). Testimonials provide qualitative evidence of the impact of the Emerging Leader program, with numerous participants explaining that the connections and experiences made whilst on the program led to new job opportunities (Table 3). Since 2023, an anonymous

feedback survey has been sent to the participants. Whilst only two years of data are available, the average score for 'satisfaction of the variety of the program' was 86/100 in 2023 and 71/100 in 2024.


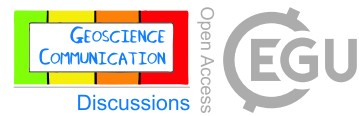

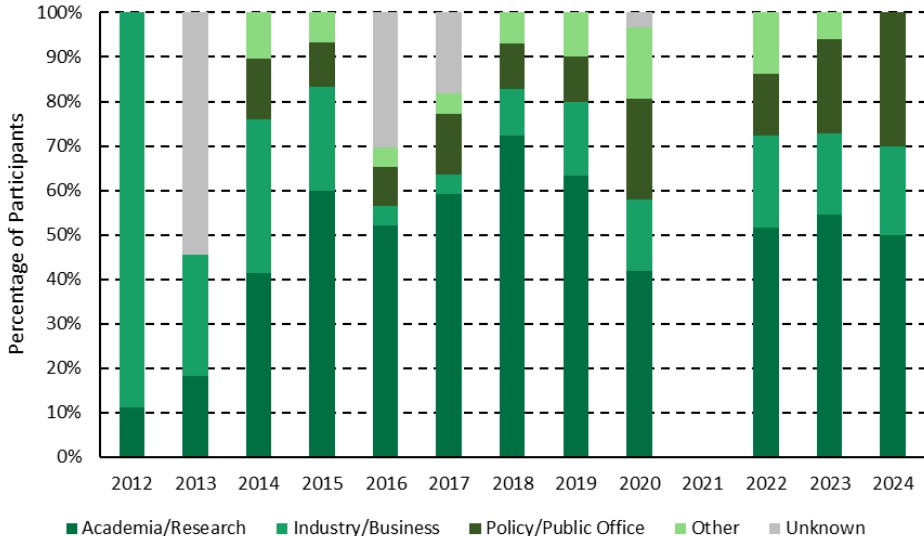

**Figure 9: The split between employment sectors within the Emerging Leader participants in each year. Academia/research includes PhD candidates.**

**Table 3: Quotes from the participants of the Student Forum and Emerging Leaders programs.**

| **Student Forum** |
|---|
| 'I got to meet all these new students from different backgrounds, different universities, giving new perspectives…gaining all this new knowledge about stressors in the Arctic.' |
| 'I think maybe my study direction is a bit clearer now.' |
| 'It has truly invigorated my passion for the topic of the Arctic and left me more ambitious than ever to tackle whatever is ahead.' |
| 'I am richer from this experience. It opened interesting collaboration opportunities, and it was really stimulating and inspiring.' |
| 'The mentoring seasons were extremely useful. Seeing the perspective from someone who is already within the world we want was a light of guidance among all the questions and uncertainty that surrounds our path already. I was also really glad that this person gave us personal advices (sic) and perspectives.' |
| 'I wish I did this earlier in my studies.' |
| **Emerging Leaders** |
| 'Very inspiring and enlightening to discuss all the work being done in other countries. This experience has brought so much inspiration, questions, and hope for the future.' |
| 'My summer experience with [job provider] would never have been possible without my participation in the Arctic Frontiers Young & Emerging Leaders program' |
| 'The week brought insightful conversation, enlightening experiences, and a hopeful view for the future. It was inspiring to |



| hear the work that's being done in other countries.' |
| :--- |
| 'It's been such a great opportunity to engage with people outside my field, it's an excellent environment to expand ones horizon and to learn and it provides a great networking opportunity.' |
| '[The program] provided insightful and new perspectives and motivated and energized me to work towards these issues in the future.' |
| 'It was very inspiring to discuss complex issues like green shift with people with so much different knowledge, expertise and perspectives. The networking aspect of EL, especially from North America has a great value and potential for future cooperation.' |

## 4 Discussion

The Arctic Frontiers Young Program fosters the next generation of scientists with a passion for Arctic issues, from the age of 5 to 35, through the four programs: Science for Kids, Science for Schools, Student Forum, and Emerging Leaders. Over 200 pre-schoolers, over 3100 high school students, 130 Bachelor and Master's students and 306 early career professionals have passed through the Arctic Frontiers Young Program since its inception in 2012. Whilst no formal monitoring of the participants has been conducted to identify a long-lasting impact of the program (e.g continuation of higher education in STEM subjects for the Science for Schools participants), qualitative surveys, LinkedIn updates and informal conversations with some previous members has resulted in positive feedback (Table 2).

The quotes of the participants (Table 2) positively support aims 1 and 3 (see Introduction), with numerous participants mentioning the impact of the program(s) on their study and job choices. Additionally, numerous participants reflect on the challenges and issues that the Arctic face, which includes climate change and ocean developments. From following participants on social media, many participants of both Student Forum and Emerging Leaders continue onto careers or further education in STEM following the programs.

For E&O activities of this scale to be successful and impactful, close collaborations with research institutes, science centres, museums, small businesses, early career networks and civil servants are required. Additionally, the numbers of participants and applications (for Student Forum and Emerging Leaders) have both increased over time, showing the importance of marketing, word of mouth and legacy for the international programs. The high levels of participation in the Science for Kids and Science for Schools activities can also partially be attributed to the high scientific literacy of Tromsø. A high number of research institutes are located in Tromsø including the Norwegian Polar Institute, UiT - The Arctic University of Norway, Tromsø, Institute of Marine Research, Norwegian Institute for Nature Research (NINA), Air Research (NILU), Water Research (NIVA) and aquaculture (Akvaplan-niva), to name a few. This results in many students with parents or family members working within the research field, and numerous science-based school and E&O projects. It also provides an ample number of experts for the Inspiration Days part of the Science for Schools program.



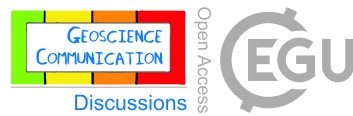

The multi-generation Young Programme is similar to the k2i (Kindergarten to Industry) approach taken by York University
in Canada in their engineering department (Cole et al. 2022). Qualitative results seen from other studies (e.g Cole et al. 2022;
Vennix et al. 2018) are reflected in the Arctic Frontiers Young Program, whereby students saw a greater understanding of
STEM and developed their professional skills early in their career.

For the two programs which require applications (Student Forum and Emerging Leaders), the last few years have been
characterised by significantly higher numbers of women than men applying to participate, and measurably higher quality
applications coming from women too. This is then reflected in the selected participants (e.g Figure 7). Similarly, despite
funding to support the inclusion of 8-10 participants from Norwegian Universities in the Emerging Leaders program, the vast
majority of those who apply are international students living in Norway, rather than Norwegian nationals. Whilst there is
limited research looking into these aspects, some parallels can be drawn from studies which identify gender and nationality
gaps in internship work, networking and mentoring programs.

Mickey et al. (2022) identified that women in the technology profession typically seek out formal networking and mentoring
events, conferences and seminars, to engage in more 'strategic networking' than men, who tend to use 'strategic socializing'
by building informal networks through social events and sports. Over-representation of women in unpaid internships
compared to men have been found in a number of studies in the USA and Canada (Hora et al. 2022), and attributed to the
normalization of low wages for the work that women do (Shade and Jacobsen, 2015). Similarly, a wage gap exists between
white American young people and first-generation immigrants or international students in the internship and apprenticeship
fields too (Hora et al. 2022). A recent journalistic investigation of the Norwegian University of Science and Technology
(NTNU) identified that just 10.8% of applicants to 43 advertised PhD positions (in the field of sustainability) were
Norwegian citizens (Vartdal, 2022). Upon interviewing the Rector of the university, Professor Tor Grande, he expressed that
many young Norwegians are employed in jobs following their bachelor's or master's degrees due to competitive labour
markets (Vartdal, 2022).

With over 4 million Indigenous Peoples in the Arctic, representing almost 9% of the Arctic's population, it is important to
include young Indigenous peoples in the E&O programs. The Student Forum and Emerging Leaders programs have both
historically included participants who self-identify as belonging to an Indigenous Community, however demographical
records are only available recently. Since 2022, Indigenous participants represented 12-13% of Emerging Leaders
participants. The Student Forum program includes students studying at the Sami University of Applied Sciences in
Guovdageaidnu/Kautokeino, Norway. No demographic data is available to monitor diversity of the children and high school
students in the two younger programs, however no specific Sami schools or kindergartens are currently included in the
program.. To ensure that harder to reach communities continue to engage in the Young Program, efforts are made to create
specific marketing and simplified application forms due to digital infrastructure difficulties in some remote locations.
Additionally, due to the wildfires in northern Canada during the 2024 Emerging Leaders application phase, a deadline
extension was provided to those coming from affected areas. To further enhance engagement, applications in different
languages could be considered, as well as beginning to monitor diversity aspects in all young programs.

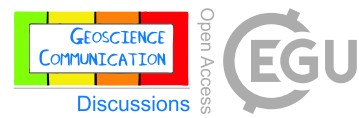

## 5 Conclusions

Here we have presented the first analysis of the four components of the Arctic Frontiers Young Program, with a focus on trends of participants (number, geographical reach, gender, demographics) over the last decade, as well as gathering feedback from the two programs for older participants, to assess the success of the projects aims. Qualitative feedback from the participants of Emerging Leaders and Student Forum points to increased interest in STEM subjects and continuing education and/or careers in STEM fields. Furthermore, the success of the program can be seen through increased applications for Emerging Leaders and Student Forum, as well as consistently high numbers of children taking part in the two earlier-years activities.

Aspects of the Arctic Frontiers Young Program have been running since 2012, which has fostered new generations of Arctic enthusiasts and scientists. With changes in company structure and staff and differing reporting requirements depending on the funding source, demographic data and monitoring of participants was patchy until recently. Going forward, demographic data and the long-term impacts of the Young Program will be collected and analysed, to further improve the diversity and inclusivity of the project. The continuation of the Young Program is heavily dependent on accessible funding. Short term funding, and funds which only cover 50% of the total costs or includes funding only for travel and project activities, but does not allocate staffing costs, are limitations to the program. Additionally, international E&O programs and the funding which supports them, are impacted by geopolitical challenges. Long-term funding for E&O programs should be provided by more sources, such as national research councils, to ensure the continued support of out-of-classroom learning experiences for young people in all education stages.

### Ethical Statement

There are no clear ethical issues with the collection of data or presentation of results. As a privately owned non-profit organisation, there are no ethical clearance guidelines. This is the first publication from Arctic Frontiers. We adhere to GDPR rules, do not name any individuals, and all participants at the education and outreach activities have signed a consent form that states that we can store and use their data.

### Acknowledgements

Arctic Frontiers is a non-profit organisation which relies on project funding for continuation of the programs. The authors would like to thank the Partner organisations of Arctic Frontiers, as well as funding from Research Council of Norway, Sparebank Nord Norge, Global Affairs Canada, Erasmus + Active Youth, the Norwegian Barents Secretariat and the work of the Science Centre of Northern Norway. The Young program has been improved and developed over many years, by different employees and especially Young Program Coordinators. The authors thank all of those who have influenced the development of the program or carried out the projects.





**Competing Interests**

The contact author has declared that none of the authors has any competing interests.

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
