# Peer review of "From Five to Thirty-Five: Fostering the Next Generation of Arctic Scientists"

_Geoscience Communication, 2024_

## Author Response (AR1)

**Dear reviewer,**

Thank you for taking your time to expertly review the manuscript and study. We have addressed your individual comments below. Author responses are in black and reviewer comments are in red italics.

It is unclear if any of the authors listed are Indigenous. In the Introduction, the authors include statistics about "lack of minority groups" in the STEM field. They do address the gender imbalance. There is a saying among Indigenous scholars, "Nothing about us, without us." In their Ethical Statement, there are a number of ethical guidelines even for privately owned non-profit organisations (see https://www.arcus.org/resources/northern-communities).

Diversity is an important aspect of all programs run by Arctic Frontiers and the authorship represents a diverse group when factoring gender, career stage, nationalities and minority groups. Arctic Frontiers has worked with numerous Arctic Indigenous peoples over the years and different programs. However, it is not the aim of any programs to directly research or address Indigenous peoples, but rather to ensure that Arctic citizens are given access to education and outreach programs. We work with Indigenous peoples from across the Pan-Arctic when advertising the programs to ensure that every voice can participate in the programs. Funding for a number of positions in the Student Forum and Emerging Leaders is ringfenced to ensure Sámi and other Indigenous groups can attend the programs without costs.

In line with many of the guidelines you shared, a long-term relationship has been developed with numerous stakeholders and rightsholders. Arctic residents (whether Indigenous or not) who are mentors or contribute to the programs are able to share their history, culture and knowledge as they see fit, and thereby influence the programs. We again reiterate that the aims of the programs are not specifically designed to increase Indigenous peoples in STEM (the same way that the aim is also not to increase woman in STEM). The aims of the programs are thus: 1) To encourage a greater number of students to take further education in STEM subjects, 2) to highlight opportunities for STEM careers within the Arctic, 3) to inspire generations of Arctic enthusiasts, who will go into science, research, policy-making and sustainable industry careers, and 4) to engage scientists in regular, impactful science communication with the public.

Increasing diversity of those accessing STEM and especially providing ring-fenced participation and funding for minority groups is an important aspect of all programs. The introduction provides a basis for why education and outreach activities such as these are necessary, in which we specifically reference minority groups due to published evidence.

Scientific significance: The four key projects sponsored by Arctic Frontiers include audiences as outlined in the authors introductions (gender and minorities): Science for Kids, Science for Schools, Student Forum, and Emerging Leaders (please note the serial comma as adopted by many writing style guides);

Thank you for pointing this out, we have amended the grammar.

Scientific quality: the authors provide quantitative data and translated qualitative data to support their article;

Presentation quality: the article is clearly written without jargon and reads well for its intended audience.

Thank you for your feedback.

Minor corrections:

"Sámi" should include the accent throughout the paper (see https://samas.no/) and Line 268 should be "...Sámi University of Applied Sciences."

We have now amended this throughout, thank you.

**Dear reviewer,**

Thank you for taking your time to expertly review the manuscript and study. We have addressed your individual comments below. Author responses are in black and reviewer comments are in red italics.

While this is an important societal topic (STEM, technical labour challenges of Arctic regions, education), and the Arctic Frontiers approach is worthwhile evaluating in this regard, in its current form the academic rigour is not up to standards to be accepted for publication. The paper generally is well written and structured. The paper contains interesting insights with regards to the various educational programmes that Arctic Frontiers has been running over the last 10 years.

Thank you for your feedback. We hope that the changes to the manuscript structure and additional methods and evaluation of the limitations of the programs can increase the academic rigour.

The key weakness of the current version of the manuscript is its lack of methodological explanation. Half-way the paper in the results section it is mentioned suddenly that the programmes are evaluated based on different types of performance criteria. What these criteria are, and how they have been determined, remains largely hidden. To be acceptable for publication the paper would require a solid methodological section in which the various steps of the evaluation methodology, the choice for particular criteria, should be clearly presented

Thank you for your feedback. The paper has undergone significant structural changes and a methodology and data analysis section are provided. Please see the full manuscript for details.

In its current form the discussion section of the paper fails to link the results of the paper to the existing literature on the topic, to conceptual choices made in the study, or to the methodology chosen for evaluation. To be acceptable for publication a more serious attempt should be made to discus the research results to clarify what is common or distinctive about these results.

More connection to existing research has now been provided, please see the full manuscript.

Page 3, like 69-71: explains the main objective of the paper ("highlight the importance of E&O projects"), which sounds more like an advocacy objective than an academic research paper objective (i.e. understand, assess, evaluate, etc). It could be that the analysis could lead to a greater appreciation of such E&O projects, but it sounds too preconceived to assume that they are before evaluation.

Thank you for your feedback, we have amended the aims to reflect the recommendations of multiple reviewers and due to the changed structure of the paper.

(Line 70-76) The aim of the paper is to evaluate the significance of continued and sustained E&O projects at all stages of education and early career development, to

provide a template and motivation for conducting Arctic-related E&O projects also outside of the Arctic, and to ensure interested and inspired scientists remain working on, and living in, the Arctic.

Since the authors of the paper seem to be strongly affiliated with Arctic Frontiers and their programmes, they would have to be very careful with mixing academic independence/neutrality and rigour with their interests in the success of their programmes. In the acknowledgement of the paper it is stated that the authors or Arctic Frontiers as a private company have no interests in the outcomes of the research. I find this not convincing at all. The authors are advised to not shy away from their interests in this issue, and to be more clear on how the methodology has enabled them to produce reliable, valid outcomes on this issue regardless. Page 3, line 76: It is mentioned that Arctic Frontiers is a private company. Please make this more clear in the introduction of the paper, what Arctic Frontiers is, what its mission is, and why as a private entity it engages with public educational institutions.

Arctic Frontiers is a private non-profit organisation, which is a partnership of the largest Norwegian institutions working towards Arctic research and challenges within the northern regions. This partnership includes numerous Norwegian public institutions, such as Norwegian Polar Institute and five major universities, who partner with Arctic Frontiers in order to use the activities and events as a way of showcasing and communicating research – a requirement of public funded research projects. Additionally, public institutions such as municipalities and county councils also provide funding and partnership with Arctic Frontiers, in order to highlight the thriving communities who live and work in the Arctic. The aim of the company is to encourage knowledge-based sustainable development of the Arctic and science/research is at the core of the events being organised and projects run. This aim can't be met without the voices of young and engaged people.

Arctic Frontiers makes no profit on the activities, and the aim of holding the extensive E+O projects is truly to encourage more young people to thrive in STEM communities within the Arctic and to see more knowledge of the Arctic integrated into programs outside of the Arctic. Our parent company, Akvaplan Niva, is also a non-profit aquaculture and water research company, who actively contribute to the academic community through their research. We hear from many organisations and researchers that they do not have the capacity to invent and innovate E+O programs, and we hope that this publication can also serve as a template for others to use. In addition, there are many private organisations whose sole mission is to provide E+O opportunities – for example APECS, International Polar Foundation, Polar Educators International.

The following is now in the manuscript, however reviewer 4 requested that less space be dedicated to describing the company, therefore we hope that the extended description above provides more information.

(Line 55-63) Arctic Frontiers is a private, non-profit organisation based in Tromsø, Norway and owned by Akvaplan-niva, an aquaculture and environmental research and consultancy company. Arctic Frontiers aims to be a catalyst for decision-making and network-building within the Arctic and consists of a partnership of over 20 public and

private institutions which include many universities and research institutes in Norway, such as the Norwegian Polar Institute, as well as private companies with interests in the Arctic. Founded in 2007, Arctic Frontiers has been gathering Arctic stakeholders from science, business, politics, and local communities for discussions regarding a range of Arctic-relevant themes, such as climate change, geopolitics, energy transition and demographic changes.

**Dear reviewer,**

Thank you for taking your time to expertly review the manuscript and study. We have addressed your individual comments below. Author responses are in black and reviewer comments are in red italics.

This paper outlines a very large outreach program run through the Arctic Frontiers organization out of Tromsø, Norway. The authors have investigated who the program reaches, and assessed whether the program is meeting its objectives. The paper reads rather like a prospectus for the outreach program and the impressive number of persons and educational institutions involved, as well as some next steps the organization is taking. There are some missed opportunities to discuss key issues in reaching local Arctic residents in these kinds of programs, in recruiting local and Indigenous researchers more generally, and what the stakes are for Arctic representation in research and policy.

Thank you for your feedback. We have now amended the structure of the paper and removed more descriptive elements to reduce the 'prospectus' feeling and ensure more analysis is included.

As the authors mention, it is a challenge securing local and Indigenous participants in the lower levels, as well as participants from Norway in the higher levels of the Arctic Frontiers program. What is the significance of this on the impacts of the Arctic Frontiers program? What is the significance of this for Arctic research more generally? The program has expanded over recent years, reaching even more participants, showing an impressive effort at outreach, but the paper could show more clearly whether this has helped the program reach their objectives. Overall, more critical reflection on the program and study could help contextualize the efforts better.

Thank you for these recommendations. We have now included more critical analysis of the programs and their limitations 'Limitations of the Young Program' section 5.

Evaluation and related amendments of the programs each year has led to improvements in some aspects of diversity and inclusion. Arctic Frontiers have increased their network by which they advertise the various programs. One method of this was through hiring people who directly work towards the young program and Emerging Leaders. They are able to invest more time in connecting with previous attendees, their own professional network and others in the Arctic community. Connecting with Norwegian Embassies overseas and applying for funding which focuses on various nations has seen an increase in diversity of those attending the programs. Applying for funding from various sources (which often has very short periods) does cause increased workload and demands on the secretariat. Furthermore, the short-lived nature of the funding calls makes it challenging to maintain focus on one nation or region of the Arctic, and therefore see long-term engagement. The program must therefore adapt each funding cycle.

The significance on research more generally is hard to say, and as we can't say if our experiences are reflected in other mentoring/youth programs in Norway or across the

Arctic. However, we do point to the NTNU study and the impacts of reduced numbers of Norwegian scientists applying for PhDs.

There are a number of dedicated Arctic research projects which actively work towards improving the inclusion of Indigenous scholars and Indigenous knowledge, for example ArcticNet in Canada. There are many oral reports of research fatigue and capacity issues for Indigenous peoples to engage with researchers. However, few of these are analysed in peer-reviewed papers. This is discussed regularly at conferences and during meetings between Indigenous leaders and researchers.

We have now provided more analysis and reflection of the programs themselves based on multiple reviewers' suggestions. Due to such large changes, we don't provide a copy of this here, but ask you to look at section 5 and 6 of the manuscript.

A suggestion might be structuring the paper more around the objectives of the program, spending less time describing the program, detailing the methods used and their limitations, and to return to the references outlined in the introduction in the discussion of the results.

Numerous reviewers recommended that we re-structure the paper to focus more on the objectives, methods and evaluation. We have now changed the structure of the manuscript and hope that the objectives and methods are clearer. A separate section with data and methods is included and the results are now structured based on the objectives of the program. We have cut down the description of the programs too. Due to the length of these changes, please see the full manuscript.

The conclusion would be improved by referring back to the objectives.

Thank you for the suggestion to restructure and focus more on the objectives. These have been included in the conclusion now too.

**Dear reviewer,**

Thank you for taking your time to expertly review the manuscript and study. The feedback and suggestions have greatly improved the substance and structure of the manuscript. We have addressed your individual comments below. Author responses are in black and reviewer comments are in red italics.

The manuscript 'From Five to Thirty-Five: Fostering the Next Generation of Arctic Scientists' aims to address the challenge of out-of-classroom education and outreach (E&O) initiatives in Arctic science by analyzing Arctic Frontiers' framework and programs since 2012. The project include four projects, targeting a wide range of audiences from children to early-career professionals. The paper is well-presented, aligned with the scope of the GC journal, and will be of benefit to readers interested in conducting similar activities in and beyond the Arctic. The paper also emphasizes the importance of raising public awareness of scientific outreach activities within Arctic and beyond, which is a valuable contribution. However, I concur with other reviewers that the manuscript currently reads like a project report rather than a research paper, and requires improvement in several areas. Below, I provide detailed feedback.

Thank you for your positive feedback on the benefit of the paper for the community. We have now amended the structure and content of the paper based on all reviewer comments and we provide details below.

**Major Comments:**

1. There needs to be a clear separation between methods, results and discussions. There is no clear method section, and some discussion content is in the results section. For example, I suggest moving some of the content from Part 3: Impact and Evaluation (e.g., lines 244, 260, and 276) into Part 4: Discussion. This includes explaining why data availability is poor, why participant numbers have decreased, and addressing the need for continued funding and marketing efforts. These points could lead to interesting discussions and provide deeper insights into the program's successes and challenges.

Based on a number of suggestions from other reviewers, the structure of the paper is now quite different, with the objectives analysed clearly, limitations of the program stated and the discussion has been amended. Due to the large changes, please see the full manuscript.

2. The data and analysis supporting the results should be strengthened, especially to bolster claims of program success. For example, the quotes from attendees used in Table 2 are minimal, which may raise concerns about the credibility and robustness of the conclusions. I recommend that the authors classify the quotes and consider applying some form of text-analysis, such as keyword frequency, to lend more rigor to the findings. Including both positive and negative feedback could make the analysis more balanced and informative.

Thank you for your comments. The quotes in the tables are a small sample of the collection, due to the limited space in the manuscript and the request of other reviewers to limit the

quotes or move them all to the supplement. In addition, many of the quotes are provided by children, who provide limited written feedback when requested – even in their mother tongue. We have now included a lengthier version of the quotes in the supplement. We have also now included a keyword frequency, as you suggested, which is also included in the results section. Thank you for making the results more robust.

3. Some of the more descriptive sections can be shortened to make them easier to read. This includes, for example, the introduction of the company. Some content, such as the detailed feedback in tables, could be moved to supplementary materials and only referred to in the manuscript text.

Thank you for your suggestions and examples. We have now removed parts of the descriptions to shorten this aspect. The introduction to the company is kept however, due to the request of reviewer 2 to understand more about the companies interests in working with public institutions.

There are now supplementary materials which further support the results. However, we do keep the tables with quotes due to the largely qualitative evaluation of the performance of the programs and how they meet their goals.

Additionally, it would be helpful to readers to see what a typical schedule or activity arrangement (as templates perhaps) looks like.

Thank you for this suggestion. We have now added a typical week schedule for Student Forum to the supplement.

Adding sub-sections to Part 3: Impact and Evaluation would enhance understanding. Sub-sections could include topics like "Factors for Success of E&O Activities" or "Gender-related Issues," allowing the reader to navigate the manuscript more easily.

As the manuscript structure has now changed quite significantly, we hope that these issues are now clearer, but please see the full manuscript.

**Minor Comments:**

1. Line 69: The manuscript highlights the importance of E&O initiatives to inspire Arctic scientists. However, it may serve a broader purpose by offering insights into conducting E&O activities in non-Arctic regions. This could be further emphasized.

The E+O activities organised by Arctic Frontiers are all taking place within the Arctic – Tromsø is located well above the Arctic Circle at 69 degrees north, and the Emerging Leaders program also begins in the Arctic. We hope that the programs can be adapted and run in non-Arctic regions, especially as outmigration from the north to the south is commonplace across the Arctic, and one of the reasons for focusing on the descriptions of the events is that it could be used as a form of template for future activities. We have now included this in the aim of the paper, and included the latitude/longitude of the regions we work in.

(Line 70-72) The aim of the paper is to highlight the importance of continued and sustained E&O projects at all stages of education and early career development, to provide a template and motivation for conducting Arctic-related E&O projects also outside of the Arctic, and to ensure interested and inspired scientists remain working on, and living in, the Arctic.

2. Line 200: The manuscript mentions that participants from over 15 countries joined in 2023. Can the authors give a specific number? Providing precise figures would increase accuracy and clarity.

We have now included a supplementary table with the list of countries involved in each year of the Emerging Leaders program.

3. Line 234: The number "550 children" is questionable, as summing the values from Figure 4 results in a higher total. Please verify this figure.

Thank you, this number has now been changed to reflect the figures.

4. Line 299: The reference to "Figures 4b and 5" may be incorrect. Is this intended to be "Figure 7b and 8" instead? Please check the numbering.

Numbers have now changed due to structural changes.

5. Part 3 (Impact and Evaluation): While the paper assesses the programs' success through participant growth and feedback, the figures show a decline in participation. I recommend explaining this discrepancy in the discussion and enriching the overall analysis.

We have now included a more in-depth analysis of how the aims of the project are being met and which factors account for success. This varies depending on the project, and increasing numbers are not always a success factor.

6. Line 316: The average satisfaction score dropped from 86/100 in 2023 to 71/100 in 2024. As a reader, I would like to know the reasons behind this decline. A deeper analysis of the factors influencing participant satisfaction is warranted.

Thank you – this has now been included. However, please note that with just two years of data (and a voluntary survey where the number who chose to respond decreased), it is hard to draw wider conclusions.

7. Line 329: The manuscript claims to assess the impact on participants' study and career choices, but this is difficult to discern from the data in Table 2. Clarifying how this table supports aims 1 and 3 would strengthen the argument.

We have now more critically reflected on the programs and how well the objectives are being met. The structure of the paper has also been amended to reflect this. Additional analysis of qualitative data is also provided. Please see the manuscript due to significant changes.

8. Line 331: The activity of following participants on social media is mentioned in the discussion without prior reference in the results section. Social media data should be introduced in Part 3 and elaborated upon in Part 4 to maintain consistency.

The activity of participants on social media is visible to the Arctic Frontiers team, but due to privacy reasons, we do not include this data in the manuscript – only prior-consented data from surveys and videos is used. Therefore, we do not include this in the data section, it is merely used as a discussion point to reflect on the career development of the groups.

**Technical Comments:**

1. Line 229: The section number should be 3.1, not 3.3.

Fixed - thank you for spotting.

2. Line 271: "Note held in 2022 due to..." should read "Not held in 2022...".

Fixed - thank you.

---

## Author Response (AR2)

**Reponse to Reviewers and Editors**

Dear Editor and Reviewers,

We appreciate your time and efforts in improving the manuscript and are thankful for the summarized version of your suggestions. Here we provide specific comments or corrections and a tracked version of the manuscript is also available. We hope that with these changes, the manuscript can be finalized and published.

**In response to Editor Summary:**

• Please revise the phrasing in the abstract and introduction to avoid referencing topics that are not substantively addressed in the paper. For example:

\*In the abstract, reconsider "to provide a template for science communication"
\*In the introduction, avoid phrasing such as "ensure... scientists remain working on, and living in, the Arctic," unless supported by data or discussion.

Thank you. These phrases have been re-written to better reflect the conclusions and focus of the paper.

• Please note also the recommendation to harmonize the introduction and discussion sections. This is an opportunity to clearly state the research questions. I recommend aligning each of the four program aims (outlined in the Introduction) with a corresponding research question. The Discussion section should then be organized to answer to these.

E.g. if Aim 1 were to be recast as What are the trends in participation across age groups, and what indications exist that these programs influence interest in STEM education? - the authors' analytical contributions address this already, and aligning the Discussion in this way will make those contributions more explicit.

Thank you for this suggestion – it has greatly improved the flow of the manuscript and much better connects the aims to the conclusions. The following research questions have been addressed:

RQ1) What are the trends in participation across age groups, countries and other demographic identifiers?, RQ2) What indications are there that the E&O programs have positively influenced the interest in STEM education?, RQ3) What evidence is there that the Young program has impacted career choices of participants and their knowledge of career options, particularly in the Arctic?

As some of the research questions can address multiple aims, we haven't assigned them that aim 1 = research question 1. For example, RQ1 addresses the number of students engaged in the programs (Aim 1) and number of scientists engaged (Aim 4). However, we do think the three research questions now address each aim and the wider impact of the programs too.

We then used A1 for the aims and RQ1 for the research questions throughout the manuscript.

• The above attention to research questions aligned with program aims, and linked organization of the Discussion section, should then be sufficient to address reviewer's points 2 (Program Description), 4 (Results: Impact), and 6 (conclusions).

Example: In comment 4 (Impact), the descriptive figures are justified when positioned as evidence in response to RQ1 (participation trends and reach). Likewise, participant feedback on interest and careers directly addresses Aim 2 (potential RQ2).

• Please address the reviewer's minor comments, request to add details on survey specifics (e.g. response rates), moving Limitations to the end of Discussion, and suggestion to remove figure 3.

Figure 3 has been removed, and the limitations section has been integrated into the discussion. The discussion has been re-structured to address the research questions. 'Limitations' section is now at the end of the discussion. Due to the large changes, we recommend reading the non-tracked version for clarity.

Final comments: Consider renaming Section 2 to better reflect the paper's analytical framing, as opposed to a purely descriptive approach (to me it reads more like a Program Context and Design section). Ensure Section 4 includes the term "Results" in the heading to clarify its role in presenting findings.

Thank you. We have called section 2 'Program Design' now following your suggestion. Section 4 now has the heading 'Results' and is then split by research question for clarity and structure.

**Specific Feedback to Reviewer 1**

General comment: the authors have done a commendable effort to showcase the research aspects of assessing the success of the Arctic Frontiers Young Program. The paper is much improved, and with a some final corrections I believe the paper will be a good feature in GC. The comments below concern mostly the harmonization of the paper, and the tables and figures. They are intended to help the authors publish an academic paper, rather than a program overview, and so I hope these comments can be helpful to the authors.

Thank you for your time and great recommendations. We have tried to incorporate as many as possible and the paper is now significantly restructured. Therefore, we recommend reading the non-tracked changed version for ease.

**1. Abstract and introduction:**

These sections still seem a little disjointed from the survey results and analysis, since the survey does not evaluate the impact of the program on many of the issues raised, e.g. the double leaky pipeline, women in higher level academic positions, and retention of teachers in the Arctic. The paper also does not provide a template for communication. The paper does provide much detail on the program itself, which begs the question of how other such programs operate, and their comparative success. The paper does raise this issue in the discussion on the limitations of the program, but this adds to the disjointed nature of the paper. Is the paper about what the introduction says or about what the discussion brings up? I recommend harmonizing the introduction and discussion sections.

Thank you for the suggestions. The details of the program were cut down significantly based on the feedback from reviewers in the first round of reviews. The word 'template' was not intended to be read as a direct translation, but rather a blue print/outline/concrete ideas which others could use to develop their own programs. We have now removed this term and changed the abstract/introduction to better outline what the aims of the paper are. However, we believe the context for conducting STEM E&O programs (to have a bottom up approach to improve some of the issues in STEM) should still remain in the introduction. We have bolstered the discussion and conclusion with more connection to the larger STEM issues too.

**2. Program description**

This section has been tightened and reads better, but it would be helpful to include here data on the participants over the years as part of the description of the program. For instance, the number of participants each year, the map of the countries of residence, but also information about their nationality, sex, etc. This information is currently in the impact section, but the authors may wish to consider moving that information here. Suggest to take out figure 3 as it can be described in one sentence.

Figure 3 has been removed, however we have now re-framed the results and discussion to answer specific research questions, therefore the other data remains in the results section.

**3. Data and methods**

Please also mention how the survey was structured and how it was administered to each group (email, LinkedIn, on paper, in person, zoom), how confidentiality was ensured, how many people responded to surveys, how many people were interviewed (including the rector), and information about their sex, nationality (other demographics), and the IRB process. Was there an effort made to ensure a representative sample of respondents? Will the survey questions be included in an appendix?

As discussed in the limitations of the program, the survey was not intended to collect data for a scientific publication, it was rather intended as a way to monitor the programs and ensure that we were evaluating and improving the services. In addition, the surveys have changed over time when there were new requests for information (e.g from funding agencies). We do not collect the demographic data of many participants – especially the younger ones. The demographic data from the other participants is also removed from the surveys and they are anonymous. This was a request of most participants, who were not willing to answer questions about their gender or other

demographics. Therefore, the information on demographics (figures in the main paper) are split from the qualitative data. We have provided more information about the survey collection in the data/methods section.

**4. Impact of the young program**

The figures and tables are a bit jumbled relative to the text that discusses them, and I recommend placing them immediately following the relevant paragraph. As it is, the text is difficult to navigate, between deleted text and figures/tables at long distances from their reference text. Generally, the figures counting participants per year, and their nationality seem less of a result than a description of the program, and I would have recommended placing them in the program description, saving this section for the survey results about the program itself and its outcomes. Regarding the connection between the introduction and the discussion, the survey seems to have asked more about how the participants experienced the program, rather than following how the program impacts retention and career choice, and so more related to the discussion on the limitations of the program than to the introduction. For instance, at Line 335, you say that "Students and teachers were not directly asked what impact the program has had on theireducational path," and therefore does not help the authors speak to the topics introduced in the introduction (but they do help vis a vis the discussion on educational programs).

Thank you for your suggestions. We have now amended the introduction and research questions for the paper. Due to the many structural changes across the multiple review stages, I recommend looking at the non-tracked changes version, so that the figures are in the correct place. Hopefully, following type-setting, they will also be formatted into the correct location.

**5. Limitations of the study**

Should this section more helpfully be called discussion? The paper currently lacks a dedicated discussion section to tie together the arguments, but this section nevertheless contains many elements of a discussion.

This section has now been tied into the discussion and the discussion has been restructured following reviewer and editor suggestions. Thank you.

This section is useful and brings many good points relevant to the results just presented. However, it is not tied well to the introduction, and the citations are very different (except Cole and Vennix). I suggest needing to harmonize the introduction and discussion so that the paper is also harmonized. The survey has not gathered data that helps develop understanding of outcomes for the issues raised in the introduction, but it may still be relevant for the discussion. For instance, the authors may want to consider including a discussion of how future surveys can be structured, or how alumni might be engaged, or even how other statistical sources (e.g. SSB) could be used to explore the program's impact. The program has been running for 12 years so there might be some evidence.

Thank you for these suggestions. We have now included a paragraph in the discussion which discusses future data collection plans and we have created better connection between the introduction and discussion.

A few key questions seem important to come back to in this section. First, why there are so many women participants but so few women in higher academic positions; second, why the program has been so successful at recruiting Indigenous participants (higher than percent of residents); third, why and with what consequences the majority of participants are not nationals but residents from other nations; and fourth, how impacts on studies and careers can be monitored. These issues merit some more discussion here.

Thank you for raising these important yet complicated questions. We have attempted to investigate them further in the discussion. We explained how we are able to include Indigenous participation so readily, identified studies to shed light on the gender results and also looked more into the citizenship vs residency result. These can be found in various parts of the discussion.

**6. Conclusion**

The conclusion indicates the paper has been an assessment of the program. Please tie this back to the introduction where the paper is also about the wider issue of STEM in the Arctic and how the program contributes.

The conclusion has now been improved following your feedback – thank you. It has changed significantly, so we recommend reading the non-tracked version for ease.

**Ethical statement**

Did the authors secure IRB approval? The authors also raise Indigenous knowledge. How does the program engage with this topic?

No, we did not secure IRB approval – in Norway this is not a formal requirement and ethics approval appears connected to health and medical research. There are a number of national committees for various topics of research, but these are advisory committees and they do not grant approval. Research institutions have overall responsibility for ethics, and some Norwegian universities have their own ethics boards, however the authors are not affiliated to such institutions. More information is available here: <a href="https://www.forskningsetikk.no/en/resources/the-research-ethics-magazine/2022-2/many-approve-the-research-themselves/">https://www.forskningsetikk.no/en/resources/the-research-ethics-magazine/2022-2/many-approve-the-research-themselves/</a>.

Despite this, we make every effort to adhere to GDPR regulations, provide clear, prior information to the participants, and their demographic data is not attributed to their survey responses. If participants chose to use their name or are included in videos or photographs, they sign a form which provides us permission to use this information. If they want to remove their participation from videos, text etc. this is possible and we remove their information. Photos and feedback from children is provided following the receipt of parental consent forms. We have included more information in the ethical statement and in the paper to clarify some of these steps.

We do not research Indigenous peoples or their knowledge systems. Indigenous peoples are included as mentors or participants in various programs (information now included in the paper), however the programs are not targeted towards Indigenous peoples as a specific group (rather, all relevant and interested young people). Despite this, we have collaborations with various Indigenous peoples and organisations, and follow their guidelines on correct useage of language and Indigenous knowledge in our programs.

Line 55 The description of Arctic Frontiers needs a sentence or two linking it to the previous sections on STEM careers, or perhaps its own heading.

Thank you – the description of Arctic Frontiers now comes immediately after the sentence calling for institutions to focus on E&O and take the burden away from scientists. Additional sentences have been added for further clarity and link to the paper.

Line 292 has a syntax issue. "Thereby..."

This sentence has been removed in the restructuring of the paper to focus more on research questions.

Figures and tables are jumbled and need to be placed as soon as possible after the referencing text so as to ease reading.

This has now been done – hopefully the layout is also easier to follow once proper proof editing and template work has been done.

---

## Author Response (AR3)

Reponse to Editor – June 2025

The current manuscript is vastly improved and has answered Reviewers' comments. Before final acceptance, please address the following to ensure clarity and consistency:

Thank you for taking time to read the manuscript and provide additional clarifications. We appreciate your quick turn-around time in this too. We have made the final changes and uploaded a new tracked version, with just the newest amendments highlighted, so that they can be seen amongst the larger changes which took place in the previous review.

• Line 18 To make clear the analytical framing (and the scope) of the study, change to: "This study evaluates the main educational activities and the best practices of the Arctic Frontiers E&O programs from the last decade, to highlight a number of possible programs which can be run in other Arctic regions." This change replaces "outlines" with the more analytical "evaluates" and specifies the focus on Arctic Frontiers E&O programs.

Changed, thank you.

• Line 23 add a more recent reference to Gibney, 2016.

We have now included a reference in the introduction and discussion. Thank you for this suggestion.

• Line 78 space needed "challengesand best practices"

Amended.

• Table 1 is too far from referencing text, it is called well-before Fig. 2 but comes after Moved under Figure 1.

Results: two organizational tasks remain:

1. Please move all reflective judgments and contextualizations (e.g. Line 263 starting with "Therefore, Science for Schools is successfully increasing" and all others e.g. line 277, 352-3, 409) to Discussion. Keep Results to only reporting statistics, but without interpretation or evaluation.

We have done this now, thank you.

2. Linked to this, remove all singposting to the Young Program's aims and move these to Discussion, to relevant points in the narrative. Section 4.3 especially heavily references aims and should be formulated in neutral terms e.g. "Quotes related to career outcomes"

also note details like Table 3 caption, revise away from aims and consider "career interest" or other appropriate term. Consider moving Line 391 "The quotes from participants reveal a positive impact" to discussion or revise: "Quotes from participants included predominantly positive language"

Thank you for specific examples and rephrasing - we have now changed these.

• Lines 235-236 Consider a more results-focused phrasing such as:

"The following section presents the results in line with the three research questions outlined in Section 1." -also note there are not four RQs.

**Changed. Thank you.**

• Line 239 section 4.1 should then start directly with "In its latest phase" in line with the above recommendations to avoid discussing aims here

**Changed. Thank you.**

• Lines 252 and 254: delete repeat calls to Figure 3 (already called on Line 248)

**Removed.**

• Line 349 section 4.2 should start with "Qualitative data from participants" as preceding sentences are interpretive (move to Discussion)

**Changed. Thank you.**

**Discussion:**

The current Discussion does well to take a broad, thematic perspective and this tone should be preserved. It is already organized in a way that can map well to a structure, it just lacks subheadings. This can be fixed easily and will greatly improve clarity without demanding a rewrite:

- Organize the Discussion section using thematic subheadings. Use revealing titles that reflect your key findings e.g. 4.1 Shifting Participation: Gender, Geography, and Inclusion. While these themes should ultimately respond to the three research questions, the structure does not need to follow the RQs directly. Please preserve your current narrative flow, but clarify it with meaningful subsection headings that guide the reader through your interpretive insights.
- As you move your connections with aims from Results to here, use simple signposting in parentheses (A1, A2), avoid fully restating aims. The task is not to rewrite the Discussion, rather to ensure the Discussion section meaningfully absorbs and reflects the interpretive statements being moved out of Results. Ensure that each of the four aims is ultimately linked to relevant findings.

We've taken these suggestions into account and clarified some sections, whilst including sub-titles which reflect the flow of the existing discussion.

Ethical statement: consider moving to Methods. ES are usually presented in the beginning

of the Methods section as a first step of data collection in participatory studies. Note that the ES switches between past and present tense, please harmonize.

The template for the layout of the publication was provided by the Geoscience Communication journal, so we have kept the Ethical Statement in the place they want it. The tenses have been changed though.

**General comments:**

- Please do a final check that all Figures, Tables and Supplementary figures and tables are called.
- Avoid repeated stating of aims and research questions beyond the introduction. Shorthand reference (RQ1, A1 etc) is sufficient.

Changes have been made. Thank you.

---

## Author Response (AR4)

Response to Editor - August 2025

Comments and edits:

(1) Remove location coordinates and the abbreviation STEM and E&O from abstract since you mention them in the text already.

Removed.

(2) Add 2-3 sentences at the end of the abstract about your core findings and the relevance of your findings to outside of the Arctic region.

The key findings have now been added. We have increased the focus on the Arctic, as one of the reviewers did not know that the programs were based inside the Arctic during previous edits. However, I have amended a sentence and set the final sentence into a broader relevance.

(3) Line 40: the authors refer to 'studies' but reference only one. Please add a few more references here.

We have included more studies, and also included additional sentences at line 53 and in the discussion section to reflect this additional literature.

(4) Line 51: non-academic research dissemination?

Changed.

(5) Line 62: Why ages between 5 and 35? Why is this significant? If it is not, why is it in the title and mentioned in the text?

It is significant due to the programs spanning such a large age range – most E&O activities are for one age group or school year, but our approach covers multiple ages, up to 35. This has now been clarified from line 76 to 80.

(6) Line 96: Please add a few references for "curriculum and pedagogy approach".

Both of the sentences in line 103 to 106 are from Rousell & Cutter-Mackenzie-Knowles, 2019 – so I have include this again. This is also stated in the Table 1 description.

(7) Why not delete 'elementary school' from Fig 1 since there is no program shown under it? Can you also add a program timeline?

We haven't amended this figure, as it is important in the international context to show that there is a gap between kindergarten and secondary school. As all four programs have different timelines

throughout the year, it is too messy to add to this figure. In addition, this information was removed from the program descriptions due to several reviewers wanting less information on the programs. Figure S5 in supplement now includes this information.

(8) Line 132: no need for the second mention of Figure 2.

Done.

(9) For section 3 (data and methods) - Can you add a table here with the following potential fields: program name, data type, timeline, participant numbers, data collection challenges, etc.? This way, the reader will immediately know what type of quantitative data are available, collected over what time, and how much of it. This can be done for both qualitative and quantitative data as one or two separate tables.

The table has now been added, in addition to the timeline graphic in the supplement. With the high number of tables and figures, we have made just one table.

(10) What about analysis of quantitative data? There's no section for it.

The reviewers and editors have made significant modifications to the structure, including removing this analysis as its own section and instead formatting the results as they are now – with both sets of data answering the three research questions. E.g figure 3 is from quantitative data. The combination of quantitative data and qualitative data to answer the research questions is the best way to analyze the aims of the programs.

(11) In section 3.3, could you explain why you use this method in relation to your research questions (as you did for your results)?

This has been added now – thank you.

(12) Line 385: It is not clear what 'greater understanding' and 'clear connection' mean - I think it's a stretch especially when the statement is based on 1 or 2 quotes.

The quotes provided in the tables are just examples, we have included all quotes in the supplement. There is just an example provided for this sentence. I've changed 'greater' to 'increased' and removed 'clear'

(13) Line 452: The authors mention 'a number of studies' but only reference one, please add a few more references.

More references included.

(14) Please remove informal language (e.g., 'don't' in line 508).

Changed.

(15) Line 509: "analysis of other indicators" - what indicators? Could you list a few for example?

These are the indicators we have looked at - so I have included the sentiment of participants as an example.

(16) Line 510: What does "many participants" mean? Please insert a number in parenthesis. Otherwise, this is very subjective. Same for anywhere else in the text where similar terms such as 'many children', 'many students', 'some students', etc. appear

These appear when we have referenced the supplementary quotes but not provided a direct number. As stated in the limitations, we have not gathered information from the surveys to directly answer these research questions, so therefore the quotes do not always relate to the specific part of the paper. By writing something like Four students reflected... it would read like we have specifically asked this question but only a few students responded, and would look like we have significantly fewer students included in the qualitative surveys. Similarly, we can't say that all students were positively impacted by our programs, because not all students responded, and not all of them gave a quote which reflects this. Therefore, there is some aspect of subjectivity in the qualitative data. This sort of writing appears most in the discussion, which is to put our findings into context with other studies, so we also don't want to repeat the results by stating 'quote 1 shows...'. I've rewritten some aspects and tried to be more concrete where possible.

---

## Author Response (AR5)

Thanks for your latest revisions. Some issues still remain. Could you please address the following points:

(1) Can you consider changing the title to "Fostering the Next Generation of Arctic Scientists"? I think it is more concise and clear. If you don't want to make this change, then perhaps you can consider "From Age 5 to 35: Fostering the Next Generation of Arctic Scientists"? I think both of these suggestions will help to draw in the readers.

**Changed.**

(2) As suggested before, please remove the abbreviation E&O in the abstract. In general, it is a good practice not to use abbreviations in the abstract.

**Changed.**

(3) Line 82 - Add (RQ) after "...the following research questions..." to indicate RQ stands for Research Questions. Same for when you state the aims (A1, A2, etc.) - not everyone is able to guess what the letters stand for. Please do this the first time these terms are introduced in the text.

**Done.**

(4) Section 3 - Data and Methods - please add a short paragraph before introducing Table 2. It's strange to start your method section with a Table. In addition, shouldn't the third column be "Timeline and activities"? Also, it would be useful to shorten the text where it is needed (e.g., For Science for Kids, the data collected could read: number of participating kindergartens, children and teachers, qualitative feedback from teachers). Since the "data collection challenges" column is more or less the same for all programs, I suggest removing it and including it in the text.

I have moved a paragraph from below the table to above it. Adding additional words would be against the advice from the previous 4 rounds of reviews where we were told to shorten the text. The data collection challenges were in the text, but you asked for them to be in a table form, which makes them more succinct, so we will leave them in the text.

(5) Line 507 - I suggest inserting (21 times in total) right after "...were common...".

**Done.**

(6) Competing Interests - I believe it is appropriate to state here that the first author is affiliated with Arctic Frontiers.

**Done.**